# Relaxation Dynamics of Chlorophyll *b* in the Sub-ps Ultrafast Timescale Measured by 2D Electronic Spectroscopy

**DOI:** 10.3390/ijms21082836

**Published:** 2020-04-18

**Authors:** Elisa Fresch, Elisabetta Collini

**Affiliations:** Department of Chemical Sciences, University of Padova, Via Marzolo 1, I-35131 Padova, Italy; elisa.fresch@phd.unipd.it

**Keywords:** chlorophyll *b*, ultrafast dynamics, 2D electronic spectroscopy, vibrational relaxation, spectral diffusion

## Abstract

A thorough characterization of the early time sub-100 fs relaxation dynamics of biologically relevant chromophores is of crucial importance for a complete understanding of the mechanisms regulating the ultrafast dynamics of the relaxation processes in more complex multichromophoric light-harvesting systems. While chlorophyll *a* has already been the object of several investigations, little has been reported on chlorophyll *b*, despite its pivotal role in many functionalities of photosynthetic proteins. Here the relaxation dynamics of chlorophyll *b* in the ultrafast regime have been characterized using 2D electronic spectroscopy. The comparison of experimental measurements performed at room temperature and 77 K allows the mechanisms and the dynamics of the sub-100 fs relaxation dynamics to be characterized, including spectral diffusion and fast internal conversion assisted by a specific set of vibrational modes.

## 1. Introduction

Natural light-harvesting systems are comprised of numerous pigment-protein complexes, consisting of different chromophores embedded in a protein matrix [1,2]. In these complexes, the light energy initially captured by pigments is delivered to the reaction centres through highly optimized energy transfer pathways. The efficiency of this machinery is heavily related to the organisation of the pigments within the protein matrix and to multiple interactions of the chromophores between each other and with the protein backbone [3,4]. The complexity of these interactions, however, often hampers a thorough comprehension of all the subtle details regulating the biological functions. Thus, preliminary investigations on the photophysical properties of isolated chromophores in different solvents may be of help to untangle their role in different environments.

In this context, several efforts have been devoted to the characterization of the steady-state and time-resolved properties of chlorophyll *a* (chl*a*), the primary photoactive pigment in biological light-harvesting (just to cite a few, see for example refs. [5,6,7,8,9,10]). Instead, less attention has been paid to another important photosynthetic pigment, chlorophyll *b* (chl*b*), whose molecular structure is reported in Figure 1a.

The biological functions of chl*b* in photosynthetic complexes have been widely studied in the literature. It is now ascertained that chl*b* expands the spectral range of chl*a* for capturing sunlight and tunnelling it to the reaction centres [11,12,13] and that it also plays an essential role in the turnover of light-harvesting complexes [14]. More recently, it has been discovered that the electronic properties of chl*b* are tuned by specific interactions between the protein backbone and the formyl group in the C7 position, whose presence distinguishes the molecular structure of chl*b* from chl*a* [15].

Fewer are the investigations on the properties of the isolated chl*b* molecule in solution and even more scarce are the works devoted to the characterisation of its relaxation dynamics. Time-resolved techniques [16,17], and in particular also multidimensional spectroscopy [18,19], have so far been exploited to unveil the ultrafast relaxation dynamics of chl*b* and identify possible variations with respect to chl*a*. These works, however, targeted the dynamics beyond the first hundreds of femtoseconds (fs), studying, for example, the spectral diffusion [18] and internal conversion dynamics [17]. Nonetheless, the very early steps of the relaxation dynamics immediately after photoexcitation are crucial to initiating all the ensuing processes, including energy transfer when the chl*b* molecule is part of a light-harvesting chromophore network.

To fill this gap, in this work we focused the attention on the sub-100 fs relaxation dynamics of chl*b* within the Q-bands, employing 2D electronic spectroscopy (2DES) with 10 fs time resolution. Chl*b* samples at room temperature (RT) and at 77 K were studied to assess the contribution of different dynamic mechanisms active in the two different conditions.

The results allowed essential details to be disclosed about the dynamics of spectral diffusion and the redistribution of energy to specific vibrational modes. Significant discrepancies with respect to the dynamics of chl*a* are also highlighted, in agreement with previous reports.

## 2. Results

Figure 1b shows the linear absorption spectra measured for chl*b* in a mixture ethanol:methanol 4:1 at room temperature (RT) and at 77 K. This particular mixture was selected for its capability of forming a good glass matrix at 77 K.

The electronic properties of chl*b*, analogously to the more studied chl*a*, can be interpreted in the framework of the well-known Gouterman’s four orbital model [20,21]. According to this model, the lowest energy transitions are labeled as Q-bands. They are the result of two partially overlapping electronic transitions identified as Q_x_ (S_0_ → S_2_) and Q_y_ (S_0_ → S_1_), with *x* and *y* indicating the polarization directions within the macrocycle plane [22]. At RT, these bands are broadened by inhomogeneous effects and by the activation of low-frequency molecular vibrations. Higher energy vibronic transitions appear instead as separated sidebands, usually identified as Q_y_(0,1) and Q_x_(0,1), respectively. In agreement with previous investigations, the lowest energy band at 650 nm (15,390 cm^−1^) is assigned to Q_y_(0,0) transition, while higher energy signals are attributed to the mixing of Q_y_(0,n) and Q_x_ states [17,22,23]. As expected [24], the spectrum at 77 K is slightly blue, shifted with respect to the one at RT, and is characterized by narrower bandwidths. These differences are also reflected in the 2DES maps shown in Figure 2.

The ultrafast dynamics of chl*b* at RT and at 77 K has been studied by means of 2DES in a fully non-colinear BOXCARS configuration [25]. The laser spectrum was tuned to cover the lowest energy part of the Q band, as depicted in Figure 1b.

Figure 2 shows the evolution of the 2DES signal as a function of the population time at RT (panel a) and 77 K (panel b). The 2DES response is cast in a series of 2D frequency-frequency maps correlating the excitation (*x* axis) and emission (*y* axis) frequencies for each value of the population time [26,27,28].

In both sets of measurements, the 2D maps are dominated by the presence of a diagonal peak that can be attributed to the ground state bleaching and stimulated emission of the Q_y_ (0,0) band excited by the laser, in analogy with what has already been reported for other chlorophylls [9,10,29,30] and tetrapyrrole compounds [31,32].

In addition to the main diagonal peak, the presence of cross peaks can also be ascertained at symmetric off-diagonal positions. In particular, the more intense cross peak below diagonal is centered at coordinates of about (16,000, 15,250) cm^−1^ and is originated by the coupling of the Q_y_ (0,0) state with vibrational modes in the range of 200–1100 cm^−1^, responsible also for the broadening and the vibronic progression in the linear absorption spectrum. Ultrafast relaxation processes from higher states can also contribute at these spectral coordinates, as verified for chl*a* [9].

A first qualitative comparison of the response at RT and 77 K immediately highlights relevant differences in the spectral shape of the signals and its associated time evolution. At RT, the 2D maps are characterized by a broader signal that is initially elongated along the diagonal and becomes more rounded as the population time increases. This behaviour is not found in the maps recorded at 77 K, where the signal appears sensibly narrower, particularly along the anti-diagonal dimension, and does not show a relevant evolution of the peak shape, at least in the investigated time window of 1 ps. The narrower bandwidth of the signal at 77 K also allows for a better resolution of the vibrational modes more strongly contributing to the lower diagonal cross peak, where signals at an excitation energy of about 15,400, 15,990 and 16,400 cm^−1^ can be identified (Figure 2b). The coordinates of these signals agree with the energy of the vibronic features clearly emerging also from the absorption spectrum at 77 K (Figure 1b) and with the frequency of the most intense vibrational modes of the chl*b* molecule, as emerging from Raman spectroscopy (see Appendix A).

To unveil more details about the dynamics of the 2D response in the two sets of measurements, the data have been analysed through a multi-exponential global fitting procedure [33,34], which allows the dynamic behaviour to be fitted along the population time at all the coordinates of the 2D maps simultaneously. It was demonstrated that this procedure could disentangle in a very efficient way the different components that contribute to the evolution of the 2DES signal [33]. The global fitting methodology was applied to the two sets of data after exclusion of the first 10 fs in order to avoid possible artefacts originating from scattering phenomena during the time overlap of the exciting pulses.

The attention was firstly focused on the non-oscillating population decay contributions. Based on the results of previous measurements [18], we were not expecting a particularly rich evolution of the non-oscillating decay signal in the first ps after photoexcitation. Indeed, the population decay can be described with a bi-exponential function both at RT and 77 K. We summarize the results of the fitting procedure in Figure 3, which reports the 2D-DAS (2D-decay associated spectra) associated with each time constant resulting from the bi-exponential fitting of both sets of measurements. A 2D-DAS map plots the amplitude distribution of each kinetic constant emerging from the fitting as a function of excitation and emission frequency; positive (negative) features in a 2D-DAS appear where the signal is exponentially decaying (rising) with that associated time constant [33].

For the solution of chl*b* at RT, the fitting analysis revealed two time constants of about 150 fs and >1 ps, respectively. The longer component captures the dynamics of all the processes that take place in a timescale longer than 1 ps [9,29,30]. These dynamics also includes the development of an excited state absorption (ESA), manifested as a negative feature at coordinates (15,410, 14,350) cm^−1^, whose intensity progressively decays as the population time increases. The same phenomenon in the same timescale has already been observed for solutions of chl*b* in different solvents [18].

The 2D-DAS associated with the first ultrafast component depicts a signal that is decaying on the diagonal (red area) and rising on the two regions above and below the diagonal (blue areas), and therefore, it captures the progressive ‘rounding’ of the main diagonal peak in the 2D maps. As already reported in the literature, when considering the dynamics of monomeric chlorophylls in different solvents, time constants of hundreds of femtoseconds are typically ascribed to the spectral diffusion process [9,10,16,18,30,35]. Indeed, the fluctuating environment surrounding an isolated chromophore changes its electronic transition frequency [10], causing a loss of correlation between the excitation and emission frequencies as the population time progresses; this is also reflected in the evolution of the shape of the main peak with the population time, as already pointed out in the comment of the 2D maps at RT (Figure 2a). The time constant of 150 fs estimated here is in good agreement with the typical timescales found for the solvation dynamics of dyes in MeOH and EtOH and assigned to the inertial component of solvation that results from the libration motion of the solvent molecules [9,10,36,37,38].

Differently from what was found for chl*a* in similar conditions, there seems to be no trace in this 2D-DAS of contributions due to the relaxation dynamics between the Q_x_ and the Q_y_ bands, manifested as a complex combination of positive and negative features at high values of excitation frequency [9]. This evidence confirms previous measurements performed with 2D electronic-vibrational spectroscopy, where signatures of ultrafast Q_x_→Q_y_ relaxation could be captured in chl*a* but not in chl*b* [19]. We cannot fully exclude that this behavior could be partially justified with a poorer overlap of the exciting laser profile with the Q_x_ of the chl*b* molecule. Nevertheless, the availability of experimental findings coming from independent measurements acquired in different conditions and with different techniques is instead a first indication of a distinctive photophysics. A possible explanation can be found in the different degree of mixing of the Q_x_ and Q_y_ bands, stronger in chl*a* than in chl*b* [23], which would also justify the modified rate of Q_x_-Q_y_ internal conversion in the two molecules [17].

A better picture of the relaxation dynamics of chl*b* in the ultrafast timescale can be achieved by analyzing the 2DES data collected at 77 K. Also for chl*b* at 77 K, the global analysis revealed two time constants of 60 fs and >1 ps. While the longer time constant describes a dynamic similar to that recorded at RT, the faster component has a shorter time constant. At 77 K it is expected that the spectral diffusion process is slowed down because the solvent fluctuations are dumped in the glass matrix formed at this temperature [39]. Therefore, it is unlikely that the 60 fs kinetics originates from a spectral diffusion process. The application of the center line slope method (CLS), a methodology often used to study spectral diffusion in 2D spectra [18,32,40], confirmed this attribution (Appendix A).

Moreover, the associated 2D-DAS exhibits a signal distribution different from the typical shape expected for spectral diffusion. Indeed, together with a decaying signal on the diagonal, it clearly shows the presence of a rising amplitude at the lower diagonal cross peak coordinates, where the vibrational modes more strongly coupled with the Q_y_ transition contribute, as already pointed out in Figure 2b.

The presence of negative features below the diagonal in the ultrafast timescale is typically associated with relaxation phenomena moving population from higher to lower energy states [9,31,41]. This relaxation can also be identified by direct inspection of the signal decay at relevant coordinates. Figure 4b shows the decay of the signal at diagonal coordinates (square) and the corresponding rise, with the same time constant, of the signal at off-diagonal coordinates (circle).

Moreover, it has been recently demonstrated that the excitation energy is quickly dumped into molecular vibrations through fast internal conversion when the modes of the bath are hindered by the low temperature [42].

These findings suggest that, at 77 K, when the spectral diffusion due to the inertial component of solvation is hindered, the first mechanism of relaxation involves the redistribution of energy to vibrational modes. Non-radiative relaxation of excited electronic states is invariably accompanied by vibrational energy redistribution, and a number of studies have already highlighted the importance of vibrational degrees of freedom in assisting the internal conversion processes in tetrapyrrole compounds [9,43,44,45,46,47]. Here, we take a step forward and, beyond providing the time constant regulating the dynamics of this process, we exploited the inherent multidimensionality of the 2DES technique to identify the main vibrational modes involved in this relaxation.

We have already discussed the advantages of the analysis of the negative signals in the 2D-DAS for the investigation of sub-100 fs relaxation processes [41]. The 2D-DAS can indeed be interpreted as a 2D frequency-frequency correlation map where the start and the final point of the energy flow are identifiable with the *x* and *y* coordinates of the negative cross-peaks, respectively [41]. The distribution of the negative amplitude signal in the 2D-DAS associated with the 60 fs time constant is characterized by an emission frequency (*y* coordinate) of about 15,000 cm^−1^, corresponding to the lowest energy Q_y_ (0,0) band and representing the final relaxed state. On the excitation axis (*x* coordinate) we recognize two maxima at about 16,000 and 17,000 cm^−1^, which suggests that vibrational relaxation mainly involves vibrational modes in the frequency range of 1000‒2000 cm^−1^. This finding is particularly relevant if compared with the results of simulations performed on chl*a* and suggests that only a subset of vibrational modes, also in the case with frequencies in the range of 1400‒2000 cm^−1^, seem to actively aid the internal conversion process [43].

The analysis of the beating behavior of the 2DES signal also supports this interpretation. The oscillating residues, obtained after the subtraction of the decaying part of the signal, have been Fourier transformed to get the so-called Fourier spectrum of coherences (FSC), which correlates the intensity of each beating component to its frequency [48]. Figure 5 reports the FSC of chl*b* at RT and at 77 K. As largely documented in the literature, the beatings in the 2DES signal of isolated dyes in solution originate from vibrational coherences in the ground and excited state, and direct correspondence with Raman spectra is typically found. This also holds for chl*b* (see Appendix A).

The overall amplitude of the beating in the chl*b* response, both at RT and 77 K, is not particularly intense. This can be justified considering the low value of the Huang-Rhys factors of the vibrational modes for chlorophyll molecules [23,24]. Nonetheless, a few vibrational modes typical of tetrapyrrole compounds and already widely documented in the literature can be identified at about 200, 296, 380, 450, 530, 770, 926, 1120 and 1260 cm^−1^ [9,29,30,35].

Moreover, it can be noticed that, with respect to RT, the amplitude of the vibrational modes with a frequency lower than 700 cm^−1^ appears strongly quenched in the FSC at 77 K (200, 296, 380, 450 and 530 cm^−1^), whereas the amplitude of higher frequency modes is more or less conserved (770, 926, 1120 and 1260 cm^−1^). This trend is somewhat expected considering that the low frequency modes are in general soft enough to be characterized by a certain degree of anharmonicity, which introduces a temperature dependence of the coupling [49] and the dephasing [50]. Nonetheless, this confirms the critical role of high frequency modes in the early time ultrafast relaxation dynamics of chl*b*.

## 3. Discussion

The sub-ps dynamics of chl*b* in an ethanol/methanol mixture has been studied by means of 2DES both at RT and 77 K. By using a global fitting analysis methodology, we could not only provide the time constants regulating the kinetics of the relaxation processes in this timescale but also get insights into the related mechanisms, thanks to the inspection of the 2D-decay associated spectra [33].

The measurements at RT allowed the characterisation of the spectral diffusion processes associated with the inertial component of solvation, to which we assigned a time constant of 150 fs. Although there has been a previous work specifically focused on the investigation of spectral diffusion in chl*b*, the time resolution of those experiments was not fast enough to clearly characterize the sub-100 fs dynamics [18]. Thus, this is the first report on chl*b* in this timescale. The comparison with the experimental results obtained on chl*a* solutions in similar experimental conditions [9] highlighted a striking difference in the internal conversion mechanism among the two molecules. Differently from chl*a*, in chl*b* there is no trace of internal conversion between Q_x_ and Q_y_ bands in the timescale of hundreds of fs. This confirms previous assumptions based on indirect experimental findings [19,43] and supports theoretical predictions suggesting a lower degree of Q_x_-Q_y_ mixing in chl*b* [23].

At 77 K, where the inertial motion of solvent molecules is frozen, the primary mechanism dominating the first stages of the relaxation of the excited state is an ultrafast (60 fs) redistribution of energy into vibrational modes. While it is known that the non-radiative relaxation of excited electronic states is invariably accompanied by vibrational energy redistribution, the detailed mechanism of how this is achieved in chl*b* was not clear. Here, besides providing a time constant for this process, we could also verify the importance of a selected subset of vibrational modes assisting the internal conversion process.

The characterization of these dynamic and mechanistic details is an important piece of information towards a better understanding of the role of chl*b* in light-harvesting complexes and we expect that these findings will be revealed to be particularly important in the future interpretation of the 2DES response of biological complexes bearing chl*b*.

## 4. Materials and Methods

Chl*b* from spinach was purchased from Sigma Aldrich and used without further purification. For the 2DES measurements at 77 K, the sample solutions were prepared by dissolving the chl*b* in a 4:1 ethanol-methanol mixture, until an optical density of about 0.3 on the maximum of the Q-bands was reached in a 0.5 mm cuvette. For the 2DES measurements at room temperature, chl*b* was dissolved in a 4:1 ethanol-methanol mixture, reaching an optical density of about 0.3 on the Q maximum in a 1 mm cuvette. We ruled out the presence of chl*b* aggregates, verifying that the normalized steady-state absorption spectrum does not change, lowering the concentration down to one order of magnitude.

The solutions of chl*b* resulted in being particularly sensitive to photooxidation. Therefore, all the solutions were immediately degassed and sealed. Steady-state absorption spectra were acquired before and after each scan to control that no degradation of the sample happened during the 2DES measurements.

2DES measurements were performed using the setup described in [25]. Briefly, the output of an 800 nm, 3kHz Ti:Sapphire laser system (Coherent Libra) was converted into a broad visible pulse in a non-collinear optical amplifier (Light Conversion TOPAS White). For these experiments the laser spectrum was centred at 15,380 cm^−1^ (650 nm) to cover the region of the Q bands, as shown in Figure 1b. The transform-limited condition for the pulses at the sample position was achieved through a prism compressor coupled with a Fastlite Dazzler pulse shaper for the fine adjustment. The pulse duration, optimized through FROG measurements, was compressed to 10 fs, corresponding to a spectral bandwidth of about 1471 cm^−1^. The pulse energy at the sample position was reduced to 7 nJ per pulse by a broadband half-waveplate/polarizer system.

The 2DES experiment relied on the passively phase-stabilized setup, where the laser output was split into four identical phase-stable beams (three exciting beams and a fourth beam further attenuated of 3 orders of magnitude and used as local oscillator, LO) in a BOXCARS geometry using a suitably designed 2D grating. Pairs of 4° CaF_2_ wedges modulated time delays between pulses. One wedge of each pair was mounted onto a translation stage that regulated the thickness of the medium crossed by the exciting beam and provided a temporal resolution of 0.07 fs. Delay times t_1_ (coherence time between first and second exciting pulse), t_2_ (population time between second and third exciting pulse) and t_3_ (rephasing time between the third exciting pulse and the emitted signal) were defined.

The outcome of the experiment was a 3D array of data describing the evolution of 2D frequency-frequency correlation maps as a function of t_2_. In each map, the excitation and emission frequency axes were obtained, Fourier transforming t_1_ and t_3_, respectively. The population time t_2_ was scanned from 0 to 1000 fs, in steps of 7.5 fs, while the coherence time t_1_ was scanned from 0 to 125 fs in steps of 3 fs. Each experiment was repeated at least five times to ensure reproducibility.

An Oxford Instruments OptistatDN cryostat was employed for the measurements at 77 K.

Data analysis was performed exploiting the global fitting methodology described in [33].

## 5. Conclusions

In conclusion, we characterized through 2DES the sub-100 fs relaxation dynamics of chl*b* in solution. The details of the mechanism regulating the excited state relaxation in this timescale were still unknown. The experiments allowed the kinetics of spectral diffusion and vibrational energy redistribution at RT and 77 K to be characterized, also highlighting significant differences with respect to the behavior of the analogous chl*a* molecules. We expect that these findings will be of great help to untangle the complex dynamics of chlorophyll molecules in biological light-harvesting complexes.

## Figures and Tables

**Figure 1 ijms-21-02836-f001:**
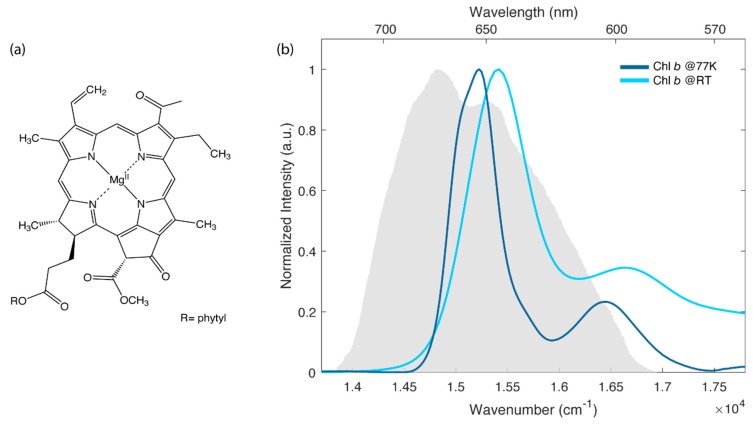
(**a**) Molecular structure of chl*b*. (**b**) Normalized absorption spectra of chl*b* in the Q bands region at room temperature (light blue) and at 77 K (dark blue). The solvent is a mixture 4:1 (v:v) of ethanol and methanol. The grey area represents the laser spectrum profile used in the 2D electronic spectroscopy (2DES) experiments.

**Figure 2 ijms-21-02836-f002:**
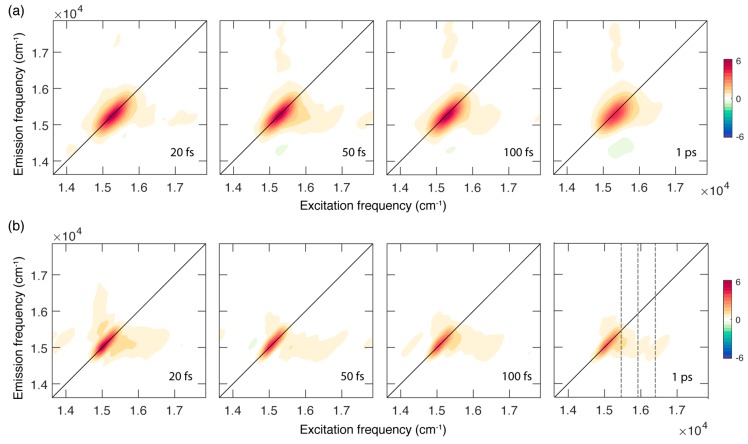
Absorptive 2DES maps of chl*b* in a mixture EtOH:MeOH 4:1 recorded at room temperature (**a**) and at 77 K (**b**) at selected values of population time. Grey dashed lines in the map at a population time of 1 ps at 77 K pinpoint the position of the main vibronic features at an excitation energy of 15,400, 15,990 and 16,400 cm^−1^.

**Figure 3 ijms-21-02836-f003:**
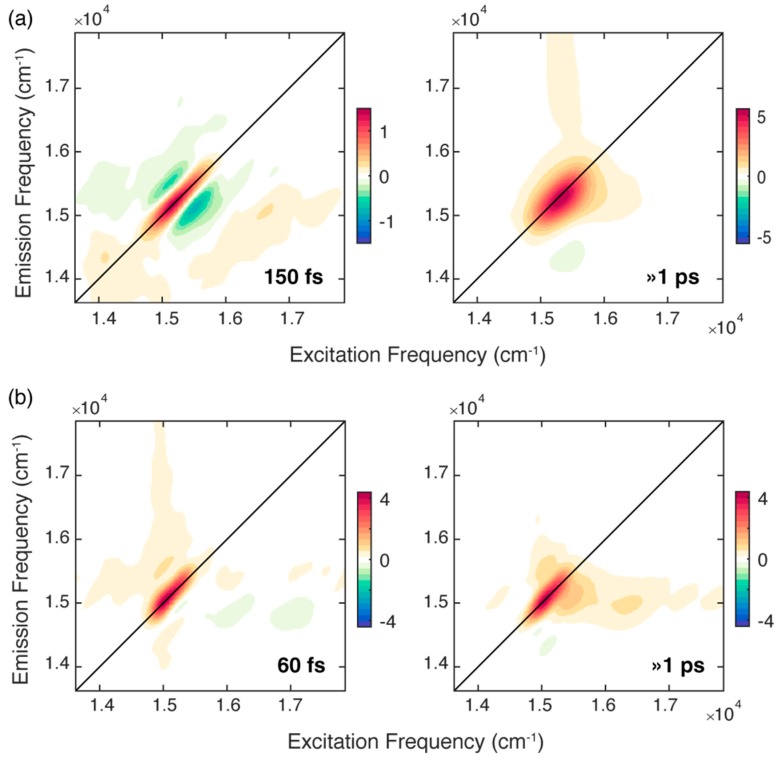
2D-decay associated spectra (2D-DAS) resulting from the global fitting of the 2DES maps recorded for chl*b* at (**a**) RT and (**b**) 77 K. The associated time constants are reported in each panel. A positive (negative) amplitude is recorded where the signal is decaying (rising) and indicated with a red (blue) colour.

**Figure 4 ijms-21-02836-f004:**
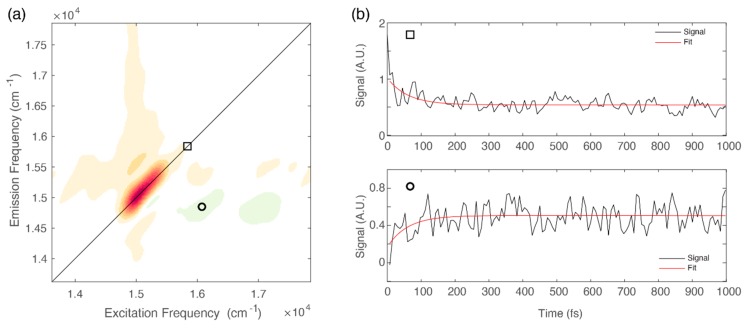
(**a**) 2D-DAS relative to the 60 fs component obtained for chl*b* at 77 K. The circle and the square markers pinpoint relevant diagonal and off-diagonal coordinates: (15,580, 15,580) and (14,900, 16,050) cm^−1^, respectively. (**b**) Signal decay extracted at relevant coordinates identified by the square (upper panel) and circle (lower panel).

**Figure 5 ijms-21-02836-f005:**
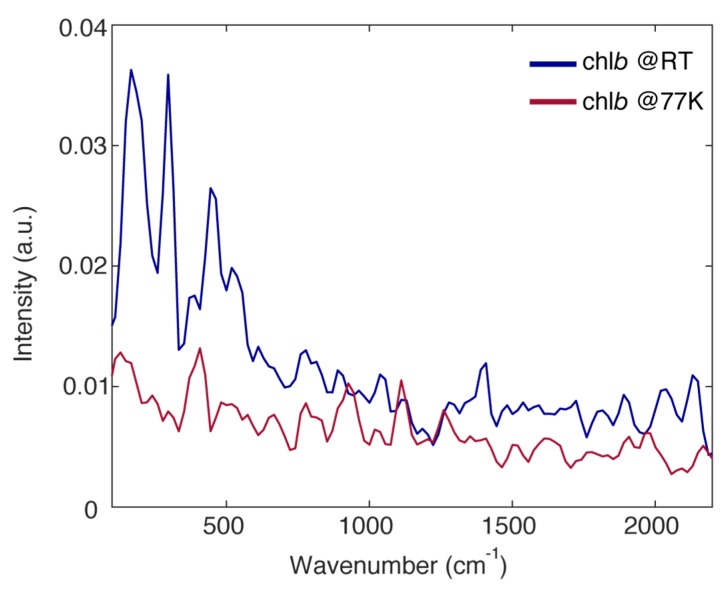
Fourier spectra of coherences obtained from the analysis of the purely absorptive 2DES maps of chl*b* at RT (blue) and at 77 K (red). Before calculating the residues, the 2DES response has been normalized on its maximum in order to allow a comparison between the amplitudes of the beatings in the two sets of measurements.

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
