# Peer review of "Relaxation Dynamics of Chlorophyll b in the Sub-ps Ultrafast Timescale Measured by 2D Electronic Spectroscopy"

_ijms, 2020, doi:10.3390/ijms21082836_

Round 1

Reviewer 1 Report

In their article, Fresch and Collini realized a detailed and systematic study on the relaxation dynamics of chlorophyll b (chlb), a naturally occurring light-harvesting pigment, in the ultrafast timescale (i.e., sub-ps). To monitor the evolution of the excited state chlb species in this early timescale, the authors used 2D electronic spectroscopy (2DES) studying the chromophore at two different temperatures, namely at room temperature and at 77K. These studies in the sub-ps time regime unveiled a different behavior in the evolution of the photoexcited chromophore as a function of the temperature, underpinning the importance of vibrational modes in assisting the internal conversion process. Moreover, the reported studies also showed, surprisingly to this referee, significant differences in the internal conversion mechanism between chlb and chlorophyll a, a structurally similar analog, something convincingly explained by the authors as due to different internal conversion between Qx and Qy bands in the two pigments in the timescale of hundreds of fs.

I reckon that the paper offers a valuable insight into an important issue as it is the understanding of the deactivation kinetics of chlb, a biologically relevant pigment, at very early timescales, a study which is unprecedented and which will definitely contribute to increase the understanding in the complex dynamics of chlorophyll molecules.

The paper is well written, the figures are clear and well-presented, and the references appropriate. I recommend the publication of this work in the International Journal of Molecular Sciences after a minor correction listed below.

  • In Figure 1, the authors should include what the R group in the molecular structure of chlb

Author Response

We thank the Referee for recognizing the value and the main message of our work.

As requested, we added the structure of the R (phytyl) group in Figure 1.

Reviewer 2 Report

The author performed 2D spectroscopic studies on Chl b at RT and at 77K, to undercover the short time ~100 fs dynamics. Chl b is an important pigment molecule in photosynthesis and deserves attention, and such studies are well needed.

After performing 2DDAS analysis, the authors found that, apart from a >>1ps dynamics, there is a 150 fs and 60 fs component, in the RT and 77K data, respectively. Qualitatively looking at the 2DDAS, the authors conclude that there is spectral diffusion at RT of ~150 fs while the features in the 77K data does not show spectral diffusion.

After addressing the following comments, it should be publishable.

line 137 – line 140: “These dynamics also includes the development of an excited state absorption (ESA), manifested as a negative feature at coordinates (15410, 14350) cm-1, whose intensity progressively increases as the population time increases. The same phenomenon in the same timescale was already observed for solutions of chlb in different solvents [18]”. This comment seems problematic, as the ESA is the absorption of the excited-state, it must exist simultaneously with the existence of the excited-state population and decrease with time when the population of the excited-state decays. In interpreting 2DDAS, amplitude, it is true that for the bleaching signal (positive amplitude in this manuscript), the positive 2D DAS amplitude represents the decays and negative amplitude the rise. However, the ESA signal has opposite sign according to the convention, so the sign of 2D DAS also has opposite meaning for the ESA signal, i.e., the negative feature of the >1-ps-2D DAS in Fig 3a indicates the decay of negative ESA signal (concomitant with the decay in the positive diagonal amplitude), not its rise.

Lines 172-173: “Moreover, the associated 2D-DAS exhibits a signal distribution different from the typical shape expected for spectral diffusion.” This comment should be verified more carefully. As spectral diffusion is a non-exponential process, the 2D peakshape evolution is non-exponential, whereas the global analysis assumes that all dynamics happening during the observed time-window is exponential. Hence, the 2D DAS do not always yield the same shape as Fig 3a for the spectral diffusion. The 2D DAS shape for spectral diffusion is highly dependent on the amplitude, timescale and even the number of components contributing into the diffusion processes. Perhaps a more quantitative analysis like using the center line slope, ellipticity or eccentricity of the 2D peakshape at 77 K to further confirm the presence or missing of spectral diffusion. Likewise these technique should also be applied to the RT data for a more quantitative analysis.

Lines 196-197: “On the excitation axis (x coordinate) we recognize two maxima at about 16000 and 17000 cm-1, which suggest that vibrational relaxation mainly involves vibrational modes in the frequency range 1000-2000 cm-1.” I agree with the authors’ comment about the meaning of the negative 2D-DAS cross-peaks. These two cross-peaks indicate the vibrational relaxation from the states at ~17000 and ~16000 wavenumber to the lowest vibrational level of Qy band. However, comparing the 2D-DAS cross-peaks coordinates and the 2D spectra, the cross-peaks rising shown in the 60-fs-2D-DAS in Fig 3b do not appear to coincide with any 2D spectral features in Fig 2b. I womder if these features are some noise compensation effects. Furthermore, the amplitudes of these features are so small compared to other peaks.

Lines 212-213:”…whereas the amplitude of higher frequency modes is more or less conserved.” It seems to me that the 77 K FSC shown in Fig 4 is rather featureless. There are no dominant vibrational modes shown up, and no obvious matches with the RT data. The authors should point out more clearly what modes are “more or less conserved”. Perhaps also show the signal before Fourier transform to prove the point,

Apart from the above, a minor error:

Line 154: “Differently from what found” lacking a ‘was’.
